# Dietary Intake, Body Composition and Performance of Professional Football Athletes in Slovenia

**DOI:** 10.3390/nu15010082

**Published:** 2022-12-24

**Authors:** Matjaž Macuh, Jana Levec, Nenad Kojić, Bojan Knap

**Affiliations:** 1Department of Food Science and Technology, Biotechnical Faculty, University of Ljubljana, Jamnikarjeva 10, 1000 Ljubljana, Slovenia; 2Department of Nephrology, University Medical Centre Ljubljana, Zaloška 7, 1000 Ljubljana, Slovenia; 3Faculty of Medicine, University of Ljubljana, Korytkova ulica 2, 1000 Ljubljana, Slovenia

**Keywords:** low energy availability, relative energy deficiency in sport, energy imbalance, sports performance, body composition measurement

## Abstract

This is the first study to examine the nutritional status of professional Slovenian football players. This study aimed to analyze the dietary intake of elite football players during their preparation phase of the season and to investigate whether there is a relationship between energy and macronutrient intake with body composition and physical performance. Twenty-three footballers completed a three-day dietary and physical activity diary and underwent body composition measurements via bioelectrical impedance vector analysis (BIVA). Fifteen participants also took part in the Cooper treadmill test to assess their physical performance in correlation with their nutritional intake. Football players had an energy intake that was significantly too low for their needs, reflecting low energy availability. The average carbohydrate (CHO) intake was below the Union of European Football Associations (UEFA) recommendations, i.e., <4 g CHO/kg body weight (BW). Additionally, players had adequate intakes of protein and fat, and inadequate intakes of saturated fat, fiber, calcium and vitamin D. There was a positive correlation between protein intake and lean body mass. Additionally, a negative correlation was observed between body fat mass and carbohydrate intake as well as between performance with the percentage of energy intake from fat. Results of this study highlight what aspects of nutrition might be improved upon in professional football players to maximize performance, longevity and body composition of athletes, as well as the necessity of a nutritionist role in this process.

## 1. Introduction

Football is currently one of the most popular sports in the world, and it places high demands on athletes’ physical capacity [1,2,3]. Professional soccer players must possess several sports related skills such as speed, endurance, mobility, flexibility, ball handling skillfulness and rapid decision-making during training and competition. In addition, the physiological demands of this sport are based on medium- to long-distance running coupled with short high-intensity movement blocks with variable patterns and shorter rest periods [3,4,5].

Football players use both anaerobic and aerobic systems during training and matches and possess unique energy and nutrient needs depending on the nature and periods of training and competition [1,6]. As such, nutrition plays an important role in optimizing performance and maintaining overall health and training longevity throughout the season. Nutrition can either potentiate or attenuate exercise-induced adaptations [7,8]. A balanced diet plays a key role in proper recovery and optimal sports performance. Changes in food intake in response to certain periods of training are also of great importance within the so-called periodization of nutrition paradigm [6,9]. Properly periodized nutrition supports performance and energy levels throughout the competition season and/or training period [7,9,10]. Nutritional requirement can usually be met with a balanced diet combined with evidence-based supplementation. Detailed guidelines on nutritional needs of elite football players can be found in the UEFA expert group statement on nutrition in elite football [7].

Despite the increased interest in nutrition and the use of nutritional supplements to improve performance, some athletes still consume a sub-optimal diet even at the elite level [11,12,13,14]. Thus, the purpose of this article was to:Evaluate the nutritional intake (energy, macronutrient and micronutrient intake) of professional football players of a football club playing in the First Slovenian League.Investigate possible correlations between dietary intake and body composition.Explore possible correlations between nutritional intake and physical performance.Present the importance of a nutritional consultant and the monitoring of nutritional intake as a method for optimizing the performance of football players and preventing relative energy deficiency syndrome (RED-S).

Based on this the following hypotheses were determined:Hypothesis 1: Football players consuming the recommended amount of protein (1.6–2.2 g/kg BW) and carbohydrates (4–8 g/kg BW) will have a body composition with a higher percentage of lean BW.Hypothesis 2: Football players who consume the recommended amount of protein (1.6–2.2 g/kg BW) and carbohydrates (4–8 g/kg BW) will cover a longer distance in the Cooper performance test.Hypothesis 3: Football players with an energy availability ≥30 kcal/kg fat free mass (FFM) will cover a longer distance in the Cooper test.

This article provides insights into the nutritional needs and dietary habits among Slovenian professional football players, their energy, macro- and micro-nutrient intakes as well as dietary supplementation habits. The results of this study offer unique insights for football players, nutritionists within the team, coaches, parents of younger players and medical staff on optimizing the performance and recovery of football players and, most importantly, protecting the long-term health of these professional athletes.

## 2. Materials and Methods

Work methods included a review of the current literature, assessment of food diaries using the Prodi program (PRODI^®^ 6.4 Expert program, Stuttgart, Deutschland), assessment of body composition and a physical fitness test using the Cooper test. Results were statistically analyzed using the appropriate statistical method and the results regarding nutritional intake were compared to the UEFA recommendations. Ethical approval was given by the Medical Ethics Commission of the Republic of Slovenia (protocol code 0120-245/2022/3, 25 July 2022). Subjects participated in the study voluntarily and could withdraw from the study at any point.

### 2.1. Subjects

25 football players aged between 18 and 35 were invited to participate in this study. All football players were members of the same club that plays in the First League of Slovenian football. Twenty-three professional football players aged between 18 and 31 responded to the invitation. This sample was chosen as we expected it to be quite homogeneous, as the footballers had a similar height and weight, level of physical fitness, training regimen as well as consuming two similar meals at the football club. The latter is a requirement of the football club were subject trained and competed at and the meals are prepared by the club’s chef.

### 2.2. Nutritional Assessment

The dietary intake of football players was assessed using 3-day food diaries (Appendix A). Diaries were provided with detailed instructions and an example of how to log in food to avoid incorrect or inaccurate logging. Subjects were asked to provide as detailed descriptions as possible of the foods and liquids consumed as well as any nutritional supplements used. Subjects were asked to pay special attention to assembled dishes at home or in restaurants (e.g., risottos, meat platters, soups) and to photograph them if possible. Additionally, subjects they were also given researchers contact information and an initiative to contact the research team if there were any questions regarding the process and food logging. Three-day food diaries were used for nutritional assessment where participants reported their eating habits on two days during the week and one day during the weekend. After completing the food diaries, each subject sat down with one of the researchers to revise the completed diaries. 

### 2.3. Physical Activity Evaluation

In addition to the food diaries participants were also instructed to fill out a physical activity diary for three days (Appendix B), in which they collected all physical activities performed during the recording of the food diaries—that is, two days during the week and one day during the weekend. Here, too participants we asked for detailed recording regarding the type of exercise they performed as well as the intensity and duration of exercise. As subjects were part of the same club their activities in training were mostly matched.

Based on these data, we calculated the metabolic equivalent (MET) value using data for individual activities from Ainsworth et al. [15]. Exercise energy expenditure (EEE) was then calculated by summation of energy expenditure of all activities. Based on these data, energy availability was calculated from energy intake analyzed through food diaries, EEE reported by football players in their physical activity diary and FFM obtained by body composition measurement using the equation by Loucks et al. [16]:$$Energy\;availability=\frac{energy\;intake\;\left(kcal\right)-EEE\;\left(kcal\right)}{FFM\;\left(kg\right)}$$

### 2.4. Body Composition Assessment

Body mass index (BMI) and height of the subjects were measured using a scale with a stadiometer. Based on this, we calculated BMI. Body composition measurements such as FFM and fat mass (FM) were obtained using BIVA (BIA 101 BIVA^®^ PRO, Class IIa Medical Device—93/42/EEC Class IIa Medical Device—93/42/EEC, Akern, Pisa, Italy), which does not provide segmental analysis and uses a three-compartment model, dividing the body into Body Cell Mass (BCM), Extra Cellular fluids and solids (ECM) and FM, where FFM = BCM + ECM.

### 2.5. Performance Assessment

Sports performance of participants was tested using the Cooper test. [17] The maximum distance participants could cover in 12 min was measured. Among 23 football players, 15 performed this test. A treadmill was used to ensure that conditions were as similar as possible for all subjects in terms of heath, wind and other weather conditions. Football players arrived at the testing site about an hour before the start to properly warm up and prepare before the test. The pace of each run was dictated but not disclosed to the individual by each subject himself, with the goal of covering the longest distance possible as done in previous research [18]. After completing the test participants could continue running at a lower intensity for a while to cool down. A qualified professional as well as doctor was also present throughout the testing, so that the process was carried out safely under expert guidance and in a safe environment.

### 2.6. Statistical Analysis

All data obtained from food diaries, body composition measurements and performance test were statistically analyzed using Excel 2016 (Microsoft Office, Redmond, WA, USA) and the statistical analysis programming language R (R version 4.1.0). Descriptive statistics were determined: minimum (min) and maximum (max) value, mean value (*x*) and standard deviation (SD). Additionally, non-parametric tests: Wilcoxon rank sum test (two-tailed test) and Welch’s *t* test (for two independent samples) were also used when applicable. To compare macronutrients intake with the UEFA recommendations a one-sided *t*-test was used to compare whether the sample averages differed from the recommendations. The normality of the distribution of individual variables and the equality of variances for independent samples were checked with the Shapiro–Wilk test. Pearson’s correlation coefficient (r) was used to examine correlations between the selected variables. Statistically significant differences between the studied groups or association between variables were confirmed if the *p*-value was less than 0.05.

## 3. Results

### 3.1. Sample Characteristics 

The sample size included 23 professional football players. The average age of the subjects was 24 years, with the youngest participant aged 18 and the oldest 31. On average, they were 182 cm tall, weighed 78 kg and had an average BMI of 23.4 kg/m^2^. They had different percentages of FFM, ranging from 66 to 86%. They also had a large range in fat mass (from 13.9 to 31%).

Through recorded daily physical activity (Appendix B), we calculated MET for each player and obtained an average value of 12.8 MET/day. Their mean EEE was 982 kcal, with a large range of 54 to 1309 kcal/day. On recorded days participants were variously active, some had rest days, while others filled in the diary during intensive multi-day training sessions. The basic characteristics of the sample are presented in Table 1.

### 3.2. Dietary Intake Evaluation

Dietary habits of football players were assessed using 3-day weighted food diaries. Data were then analyzed using the PRODI^®^ 6.4 Expert program and R statistical processing programming language (R version 4.1.0, Vienna, Austria). Energy availability was calculated using the formula by Loucks et al. [16] as explained in greater detail in the methods section. The average values of energy intake and energy availability, as well as the intake of carbohydrates, proteins, fats, saturated fats and fibers are presented in Table 2.

#### 3.2.1. Energy Availability

The normality of data distribution was tested using the Shapiro–Wilk test, which confirmed normal distribution of energy availability data (*p* > 0.05). The average energy intake was 2700 kcal and in the majority of subjects energy intake changed proportionally with a higher or lower EEE on a given day. Energy availability was calculated [9] because it gives further insights on energy balance than energy intake, as it takes into account energy intake, EEE and FFM.

Values below 30 kcal/kg FFM as clinically low energy availability, values between 30 and 40 kcal/kg FFM as subclinical low energy availability, and optimal or high energy availability as equal to or exceeding 40 kcal /kg FFM [19,20]. For women, this limit was set slightly higher, above 45 kcal/kg FFM [19]. Using a one-sided *t*-test, we found that the sample average of energy availability is statistically significantly lower than 40 kcal/kg FFM (*t* = −6.23, df = 22, *p* < 0.001). The average energy availability of the subjects was 29 kcal/kg FFM, which corresponds to the criteria of clinical energy availability (Figure 1).

#### 3.2.2. Macronutrient Intake

The Shapiro–Wilk test indicated normal distribution of data (*p* > 0.05) for carbohydrate, protein and fat intake. During the recording of food and physical activity diaries, the football players were in preseason training period. For this period, a carbohydrate intake between 4 and 8 g CHO/kg BW is recommended. The average carbohydrate intake among soccer players was 3.6 g/kg BW. A one-tailed *t*-test showed that the sample mean carbohydrate intake is statistically significantly lower than 4 g/kg BW (*t* = −2.25, df = 22, *p* = 0.017). Most football players did not reach the lower recommended UEFA lower limit of 4 g CHO/kg body weight for preseason training (Figure 2). 

Dietary intake of protein and fat was adequate among subjects. On average, they consumed 1.8 g of protein/kg BW, which is in line with the recommendations (1.6–2.2 g protein/kg BW). However, 35% of football players did not reach the lower limit of 1.6 g of protein/kg body weight (Figure 3). The range of protein consumption among football players was between 0.9 to 3.5 g/kg BW. A one-tailed *t*-test showed that the sample mean protein intake is not statistically significantly higher than 2.2 g/kg BW (*t* = −3.46, df = 22, *p* = 0.99).

The percentage of fat intake varied greatly between players—from 24% to 45% of their total daily energy intake. The recommendation for fat intake is up to 35% of the daily energy intake, and the average percentage of fat intake among the subjects was 34.7%, i.e., just below the upper limit of the recommendations (Figure 4). Since fat intake is restricted only by the upper limit and every fat intake below that limit is considered in line with UEFA recommendations, we investigated only if our sample’s fat intake is higher than the recommended percentage of fat intake. Due to the inherent directionality of our investigation, we used a one-sided *t*-test, which showed that the sample average of the percentage of fat in the diet is not statistically significantly higher than 35% of the total energy intake (*t* = −0.29, df = 22, *p* = 0.62). 

#### 3.2.3. Micronutrient Intake and Supplementation

When analyzing food diaries, nutritional supplements of various macro- and micronutrients were considered. In Table 3 and Figure 5, the results of the analysis of food diaries and a comparison with Slovenian (National Institute of Public Health) NIJZ reference values [21] and UEFA nutritional recommendations [7] are presented.

Iron, magnesium, zinc, folate, vitamins C, B2, B6 and B12 and E were consumed by the subjects to a sufficient extent; however, subjects exceeded the upper limit for salt intake for the general population (Figure 5). Participants calcium intake was below the reference value with a mean value of 964 mg/day. Nine subjects (<40% participants) had a calcium intake above 1000 mg and only one subject had a calcium intake higher than 1500 mg as recommended in the UEFA dietary guidelines for football players with low energy availability [7]. Another micronutrient that should be paid attention to is vitamin D. Participants did not cover their needs for this vitamin with their diet, only two of the participants (in the early spring months, when the endogenous synthesis of vitamin D is insufficient) took a vitamin D supplement. Football players should be advised to take regular blood tests for vitamin D status and, in case of a deficiency, supplement accordingly.

Participants most often supplemented their diet with magnesium, vitamin B6 and zinc (via food supplement ZMA), even though there was no need for this as they had already ingested enough of these micronutrients through diet alone. Certain players also supplemented with protein powder and multivitamins. The football club made and isotonic drink available during training and a protein drink with a banana and creatine after training. One subject supplemented with omega-3 fatty acids, iron and vitamin C. Some players consumed branched-chain amino acids (BCAA), L-carnitine and a mass gainer. No one used beta-alanine, nitrates, or sodium bicarbonate as an ergogenic aid even though these supplements might be useful in this type of sport and have high level of evidence [22].

### 3.3. Comparison of Absolute Values of Dietary Intake and Body Composition

The Shapiro–Wilk test showed a normal distribution of data (*p* > 0.05) for body fat percentage and FFM. Further correlations were determined using Pearson’s correlation coefficient (r).

#### 3.3.1. Energy Availability and Body Composition

The correlation between energy availability and body composition was analyzed using the Pearson’s correlation coefficient (r). We observed the effect of the percentage of body fat and lean BMI. Figure 6 and Figure 7 show no correlation between the variables (*p* > 0.05). 

#### 3.3.2. Carbohydrate Intake and Body Composition

Pearson’s correlation coefficient (r) was used to determine the association between carbohydrate intake and the percentage of body fat and lean BM. Carbohydrate intake and fat mass percentage were shown to have moderate negative correlation (r = −0.45, *t* = −2.2959, df = 21, *p* = 0.032) (Figure 8). There was no correlation (*p* > 0.05) between carbohydrate intake and percentage of lean body mass (Figure 9).

#### 3.3.3. Protein Intake and Body Composition

Pearson correlation coefficient (r) was used to examine the association between protein intake and the percentage of body fat and lean BM. We found no correlation (*p* > 0.05) between protein intake and body fat percentage (Figure 10). We found a moderate positive correlation between protein intake and lean body mass percentage (r = 0.39, *t* = 2.0905, df = 21, *p* = 0.04892) (Figure 11).

#### 3.3.4. Fat Intake and Body Composition

We investigated the influence of the percentage of energy intake from fat on fat and lean BM. Using Pearson’s correlation coefficient (r) showed no correlation (*p* > 0.05) between fat intake and fat or lean BM (Figure 12 and Figure 13).

### 3.4. Comparison of Absolute Values of Food Intake and Physical Performance

The Shapiro–Wilk test showed a normal distribution of data (*p* > 0.05) in the Cooper performance test. Further correlations were determined using Pearson’s correlation coefficient (r).

#### 3.4.1. Energy Availability and Physical Performance

Using Pearson’s correlation coefficient (r), we found no correlation (*p* > 0.05) between energy availability and the distance covered in the Cooper test (Figure 14).

#### 3.4.2. Carbohydrate Intake and Physical Performance

Using Pearson correlation coefficient (r), we found no correlation (*p* > 0.05) between carbohydrate intake and the distance covered in the Cooper test (Figure 15).

#### 3.4.3. Protein Intake and Physical Performance

Using Pearson correlation coefficient (r) we found no correlation (*p* > 0.05) between protein intake and the distance covered in the Cooper test (Figure 16).

#### 3.4.4. Fat Intake and Physical Performance

Using Pearson correlation coefficient (r), we found a strong negative correlation between the percentage of fat in the diet and the distance covered in the Cooper test (r = −0.54, *t* = −2.3216, df = 13, *p* = 0.037) (Figure 17).

### 3.5. Results of Hypothesis Testing

For each test the subjects were divided into two groups. When subjects were divided according to UEFA recommendations, depending on the training period, Group A consisted of those football players consuming appropriate intake of protein (1.6–2.2 g/kg BW) and carbohydrates (4–8 g/kg BW), while Group B consisted of players consuming inappropriate intake of protein (<1.6 or >2.2 g/kg BW) or carbohydrates (<4 or >8 g/kg BW). Thus, 8 out of 23 subjects were designated into group A, and 15 were designated to group B.

When subjects were divided according to energy availability, Group A consisted of those who had an energy availability >30 kcal/kg FFM, while Group B consisted of those who had clinically low energy availability (<30 kcal/kg FFM). Out of 15 subjects with available data, 9 were designated into group A and 6 were designated to group B.

#### 3.5.1. Hypothesis 1

Hypothesis 1 proposes that football players consuming the recommended amount of protein (1.6–2.2 g/kg BW) and carbohydrates (4–8 g/kg BW), depending on the training period, will have a body composition with a higher percentage lean body mass.

Football players in group A had an average percentage of lean body mass of 77.9%, with a minimum percentage of 72.8% and a maximum percentage of 86.1%. The average percentage of lean body mass in group B was 75.7%, with a minimum share of 66.3% and a maximum share of 79.6% (Figure 18). Hypothesis 1 was rejected in favor of the null hypothesis using the Wilcoxon rank sum test (two-tailed test). We found no statistically significant differences in the percentage of lean body mass between the two groups (W = 54, *p* = 0.7284).

#### 3.5.2. Hypothesis 2

Hypothesis 2 states that football players who consumed the recommended amount of protein (1.6–2.2 g/kg BW) and carbohydrates (4–8 g/kg BW), depending on the training period, will cover a longer distance in the Cooper test.

15 football players took part in the Cooper performance test. There were 5 football players from group A, and they covered an average distance of 3.06 km in 12 min, with a minimum distance of 2.56 km and a maximum distance of 3.5 km. The average distance covered in group B was 300 m shorter (2.76 km), with a minimum distance of 2.4 km and a maximum distance of 2.95 km (Figure 19). Using Welch’s *t*-test we rejected hypothesis 2, as we found no statistically significant differences between the two groups in the distance covered in the Cooper test (*t* = 1.8255, df = 4.8542, *p* = 0.1293).

#### 3.5.3. Hypothesis 3

In the third hypothesis we assumed that football players with an energy availability ≥30 kcal/kg FFM will cover a longer distance on the Cooper test than those with a clinically low energy availability <30 kcal/kg FFM. To test this hypothesis, we divided the subjects into two new groups. Group A consisted of those who had an energy availability >30 kcal/kg FFM. Group B consisted of those who had clinically low energy availability (<30 kcal/kg FFM).

Initially, we divided the football players into a group with a threshold value of 40 kcal/kg FFM, as this is supposed to be the energy availability that is optimal for their performance, recovery and long-term health. However, due to low sample size such a group would only consist of two football players, which was not enough for statistical analysis. Therefore, we moved the threshold value one level lower, to 30 kcal/kg FFM, which represents the border between subclinical (≥30 kcal/kg FFM) and clinical (<30 kcal/kg FFM) low energy availability [20].

There were 9 soccer players in group A and 6 in group B. The average distance covered in 12 min in group A was 2.91 km, and in group B with low energy availability, it was 2.8 km (Figure 20). With the Welch’s *t* test, we rejected hypothesis 3 in favor of the null hypothesis. We found no statistically significant differences between the two groups in the distance traveled on the Cooper test (*t* = 0.94855, df = 10.051, *p* = 0.3651).

## 4. Discussion

This is the first study looking into the nutritional status of professional football players in Slovenia. The results of this study indicate similar results as those seen in previous research, namely that football players did not meet their energy and carbohydrate needs [1,4,7,22]. Football players participating in this study had low energy availability (mean energy availability was below 40 kcal/kg FFM). Mean carbohydrate intake was 3.6 g CHO/kg BW, which is below the recommended amount of 4–8 g CHO/kg BW for preseason training [7]. Previous researchers reported average intakes of 2.4 g CHO/kg BW per day among 46 professional Australian football players [23], while Książek et al. [10] reported an average intake of 5.1 g CHO/kg BW in professional Polish football players. A meta-analysis by Steffl et al. [1] found that the mean carbohydrate intake among adult soccer players was 4.7 g/kg BW. The reasons for such a low carbohydrate intake are difficult to identify; however, when talking to participants in this study, it seems they are under the impression that carbohydrates lead to fat gain and that their intake should be limited. Football players must be made aware that the recommendations for a balanced diet for the general population are not the same as the recommendations for sports nutrition, especially for elite athletes [24].

Protein intake was found to be adequate and within guidelines (1.6–2.2 g/kg BW) with fat intake being just below the upper recommended limit of 35% total energy intake. A low intake of fiber (mean value 27.4 g) and an excessively high intake of saturated fatty acids (10.5% of daily energy intake) as well as salt (mean value 7.6 g) were reported. Fiber intake needs special attention in an athlete’s diet, and it is recommended to limit it before, during and after physical activity to avoid digestive issues. Nevertheless, athletes would also be advised to have a sufficient intake of fiber, as this is consistently associated with a variety of beneficial health effects [25]. Despite having to limit fiber intake before, during and after training, sufficient intake of fiber for football players should not be difficult due to increased daily energy intake. Perhaps the low fiber intake found in this study could be explained by overall low carbohydrate intake. The intake of saturated fatty acids should also be given attention and slightly reduced so that it does not exceed 10% of the daily energy intake. In many studies, excessive intake of saturated fatty acids is associated with negative health effects, to which even athletes are not immune. Regarding sodium and salt intake, the recommendation for athletes differs from the recommendation for the general population. Athletes lose large amounts of sodium through sweat, and replacing this mineral is key for rehydration purposes especially before, during and after physical activity. It is impossible to give a uniform recommendation for the appropriate salt intake for athletes, as the amount of sodium loss varies depending on the individual’s sweat composition, the amount, intensity and duration of physical activity and other external factors (e.g., temperature and humidity) [26]. The intake of vitamins and minerals was adequate among subjects; only the intake levels of calcium and vitamin D were found to be too low. The average daily intake of calcium was 964.2 mg/day, i.e., below the recommendations (1000 mg/day) in 60.9% of the players. The average vitamin D intake was 4.1 µg/day, including vitamin D supplementation, which is also below the recommendations (20 µg/day) in all players, and this may negatively impact immune and musculoskeletal functions [27]. Similar results regarding fiber and calcium intake were also found in Jenner et al. [23] where fiber intake among Australian professional footballers was 27 g/day and calcium intake was 952 mg/day.

Energy needs of football players are greatly increased due to the high volume and intensity of training and matches. Football players need energy for optimal sports performance, as well as optimal recovery after activity and to support all other body functions and long-term health [7]. Calculating energy availability gives insights whether an athlete consumes enough energy to support the above-mentioned processes. Energy availability should be higher than 40 kcal/kg FFM for athletes for optimal functioning and no lower than 30 kcal/kg FFM [20]. Our sample of 23 football players had an average energy availability statistically significantly lower than 40 kcal/kg FFM. Low energy availability is an important risk factor for health complications such as a weakened immune system, lower bone density, increased risk of fractures, hormonal imbalances, fatigue and depression. At the same time, low energy intake may impair performance. Based on these data, we formulated our third hypothesis, which states that football players with an energy availability ≥30 kcal/kg FFM will cover a longer distance on the Cooper test than those with clinically low energy availability <30 kcal/kg FFM. The results of this study, however, do not support this hypothesis, possibly due to low sample size. It is possible that a larger sample size would produce different results. Considering, that only 2 out of 23 soccer players had an energy availability above 40 kcal/kg FFM, we had to lower the limit to 30 kcal/kg FFM for statistical analysis and thus managed to get two groups more similar in size, between which we compared the distance covered in the Cooper test. Our analysis thus has two major limitations, the first being that our sample as a whole had suboptimal energy availability (below 40 kcal/kg FFM) and the second being the small sample size of 15 football players.

In addition to overall energy intake, physical performance is also heavily influenced by the appropriate ratio between macronutrient intakes. As such, we opted to determine whether differences between macronutrient intake, mainly the intake of fats and carbohydrates would influence athletes’ performance in the Cooper test. A negative correlation was recorded between the percentage of energy from fat in the diet and the distance covered in the Cooper test (*p* = 0.037). However, we found no statistically significant differences in distance covered in the Cooper test between football players consuming the recommended amount of protein (1.6–2.2 g/kg BW) and carbohydrates (4–8 g/kg BW) versus players not consuming the recommended amounts of protein and carbohydrates. Reasons as to why no statistically significant difference between the groups was found might be attributed to the same study limitations as mentioned previously, as very few players covered both nutritional recommendations. 

We also examined how body composition was influenced by nutritional intake. A statistically significant negative correlation was found between carbohydrate intake and percentage of fat mass (*p* = 0.032) and a positive correlation between protein intake and percentage of lean body mass (*p* = 0.04892). However, due to possible inaccuracy of body composition measurement using BIVA [28] and possible inadequate diet reporting, we question these findings. To confirm these correlations, it would be reasonable to repeat the measurements of the football players using a more accurate technique of body composition measurements such as a dual-energy X-ray absorptiometry (DEXA) device [29]. In our first hypothesis, we assumed that football players who consume the recommended amount of protein (1.6–2.2 g/kg BW) and carbohydrates (4–8 g/kg BW), depending on the training period, will have a body composition with a higher percentage of lean BM. We rejected the hypothesis, as there were no statistically significant differences in the percentage of lean BM between the groups with appropriate and inappropriate intakes of macronutrients. We must highlight the fact that our results of body composition analysis with BIVA are quite different from the results of Jenner et al. [23], who measured the body composition of Australian professional football players using a DEXA. Comparing body composition of athletes in this study the average BM of football players in this study was 78 kg, while the sample of Jenner et al. [23] was 86.3 kg. The average fat mass of football players in this study was 23.1%, while the sample of Jenner et al. [30] was 10.8%, and the average lean BM of football players in this study was 59.4 kg compared to 73.9 kg in the study by Jenner et al. [23]. Proper comparison might again be limited due to body composition measurement limitations [28].

The results of this study indicate that nutritional intake in the studied sample was not adequate and did not meet recommended UEFA guidelines [7]. A mayor concern was a notable bottom-down hierarchy approach seen in athletes regarding their nutrition, i.e., overlooking the basics (appropriate energy intake and macronutrient intake) and using unwarranted supplements. A prominent example is widespread usage of vitamin B6, which athletes already ingested enough through food alone. Using supplements without evidence base or objective needs (e.g., nutritional anamnesis and/or bloodwork) is at best an unnecessary waste of money, and at worst may have major negative consequences, e.g., excessive intake of vitamin B6 can cause neuropathy [30]. 

Another limitation of this study is nutritional tracking. Precise instructions and nutritional support as well as a visual material of different foods and meals in relation to their mass in grams were given to all football players to reduce the risk of improperly completed food diaries. Nevertheless, possible limitations of our study include reliability and validity of the estimation of average dietary intake, which may have been based on the subjective evaluations of the subjects. Some may have under-reported, and others may have over-reported intake of certain foods. In addition, the inaccuracy of the nutritional analysis may also be the result of searching for suitable foods in the Prodi program, where it was not always possible to find an identical product. A special challenge was the entry of assembled dishes (e.g., cheese dumplings) or restaurant dishes (e.g., chicken risotto), where there could be a large discrepancy between the recipes themselves. In the program, it was also necessary to pay attention to certain foods that did not have information on all nutrients, and an error in the sum of nutrients could also occur because of this. Regarding food tracking, participants eat quite differently despite having two meals provided for them at the football club. At breakfast, they were given a wide variety of different foods to choose from. The foods provided were the following: eggs, cereal, oats, a wide variety of fruits, regular and Greek yogurt, milk, butter, marmalade, bread, coffee, fruit juice, and different types of salami and cheese. Subjects could choose from any of these foods and quantity of these foods of their own volition. Consequently, subjects eat quite differently at this meal. The other meal provided was lunch, the most uniform meal across all subjects, differing slightly only in quantities of carbohydrates eaten in the form of pasta, rice, potatoes, etc. However, the most considerable discrepancies seen in nutritional intakes were seen after lunch when participants had to provide their own meals. Afternoon snacks and dinners varied greatly between individuals, and this could explain why nutritional intakes (Figure 1, Figure 2, Figure 3 and Figure 4) were recorded very differently.

As stated previously, we found no correlations for certain parameters, which may have been due to low sample size (*n* = 23 or *n* = 15 in the case of performance tests) considering the effect size obtained. To check the correlations, we chose the Pearson correlation coefficient (r), where the statistical power is also reduced due to the smaller sample. Perhaps some correlations could have been shown if more football players had been included in the research. The sample of subjects is indeed homogeneous, but still small, and when measuring performance, it is further reduced by eight players.

The limiting factor of our research was also the available equipment. Body composition measurements were performed with the BIVA device, which has limitations in giving accurate body composition measurements and where results might have been influenced by a variety of factors, namely players hydration status [28]. This inaccuracy is probably the reason why the percentages of fat mass among football players were quite high (the average percentage of fat mass among football players was 23.1%—i.e., higher than the percentages of fat mass found in the general population, which we deem impossible). In addition, the Cooper test was measured indoors on a treadmill. We decided to do this because we wanted to exclude any disturbing external factors (wind, rain, cold, heat); however, new restrictions appeared at the same time. The treadmill had a certain maximum speed, which two of the fastest football players could exceed and, as a result, might have run an even longer distance if given the opportunity. 

Not a limitation per se, but important insight nonetheless is also timing of nutritional interventions. Six months post-intervention, a notable change in the football team was noted with 10 athletes being changed in this time frame due to transfers into different national and international clubs. Perhaps it is the nature of football as a sport, or this might be a bigger problem within smaller teams that many athletes deem as steppingstones towards more notable clubs and prominent leagues. Researchers wanting to examine the effects of long-term nutritional education within a football club should take this into account.

Future research should take the highlighted limitations into consideration and should investigate internal and external factors that influence the nutritional intake of athletes. It is important to find out how to present research conclusions and sports nutrition guidelines in an understandable way for the athletes, so that they can easily and quickly understand them and integrate them into their everyday life in the long term.

## 5. Conclusions

In conclusion, this study showed that football players’ protein and fat intake levels were in line with recommendations; however, carbohydrate intake was too low in regard to the preseason training period. The intake of saturated fats was slightly above the recommended amount. Average fiber intake among football players was below the recommendations. Intakes of calcium and vitamin D were also too low in most of the subjects, while the requirements for other micronutrients were successfully met and in certain cases exceeded the recommended amount via supplementation. There were no statistically significant differences between the percentage of lean body mass between the groups with appropriate and inappropriate carbohydrate and protein intake. However, a negative correlation between carbohydrate intake and fat mass and a positive correlation between protein intake and the percentage of lean body mass were observed. There were no statistically significant differences in performance between groups with appropriate and inappropriate intakes of carbohydrates and protein, nor between groups with higher or lower energy availability. Only a negative correlation between the percentage of fat in the diet and the distance covered in the Cooper test was observed. Further research with a larger sample and more accurate recording of nutritional intake and body composition measurements is needed. 

## Figures and Tables

**Figure 1 nutrients-15-00082-f001:**
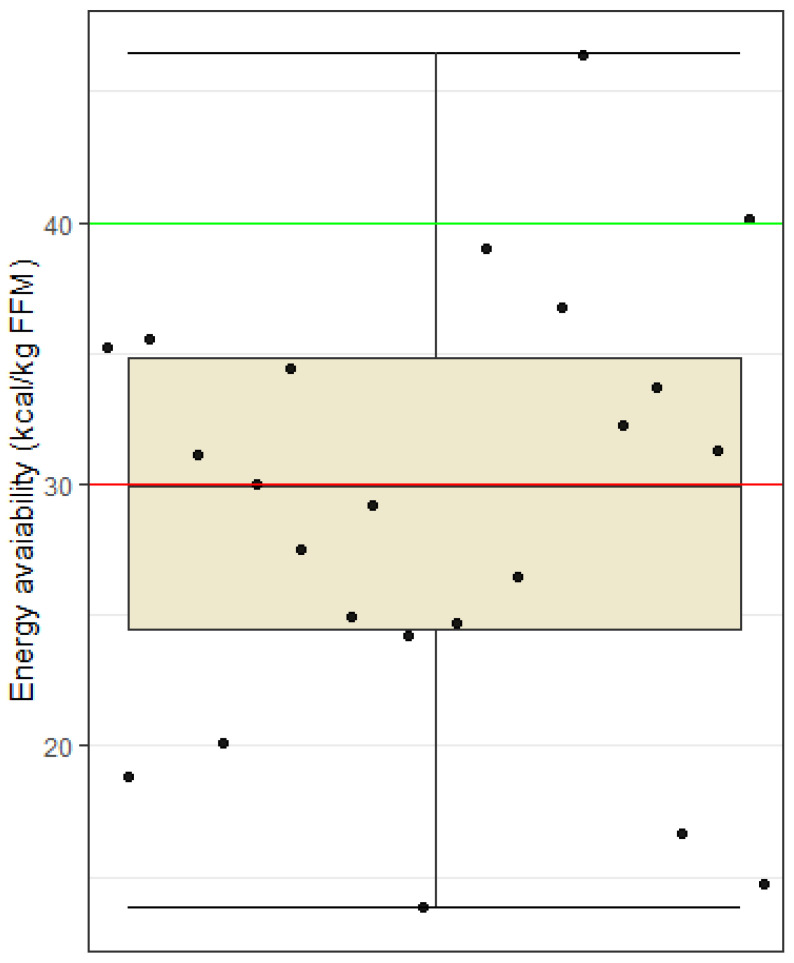
Energy availability among football players. Dots represent individual participants (*n* = 23). The red line indicates the upper limit of clinical energy availability (30 kcal/kg FFM) and the green line the lower limit of optimal energy availability (40 kcal/kg FFM).

**Figure 2 nutrients-15-00082-f002:**
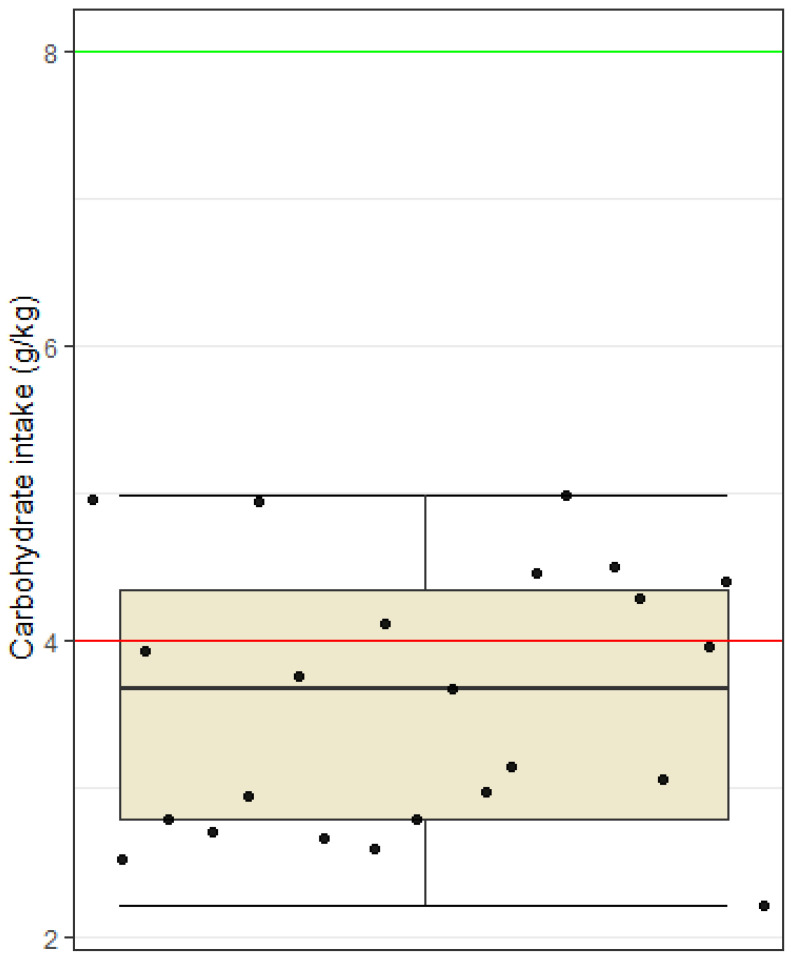
Carbohydrate intake among football players. Dots represent individual participants (*n* = 23). The red line indicates the lower limit of the recommended intake for football players in the preseason period (4 g/kg BW), and the green line indicates the upper limit of the recommendation (8 g/kg BW).

**Figure 3 nutrients-15-00082-f003:**
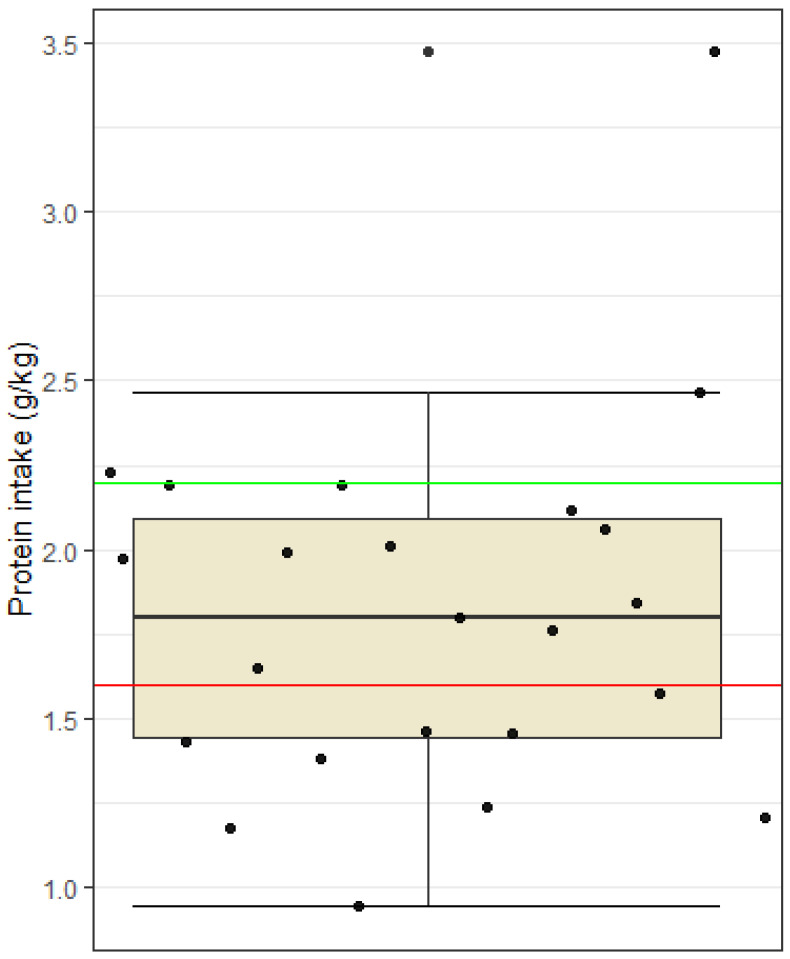
Protein intake among football players. Dots represent individual participants (*n* = 23). The red line indicates the lower limit of the recommended intake (1.6 g/kg BW) and the green line the upper limit of the recommendations (2.2 g/kg BW).

**Figure 4 nutrients-15-00082-f004:**
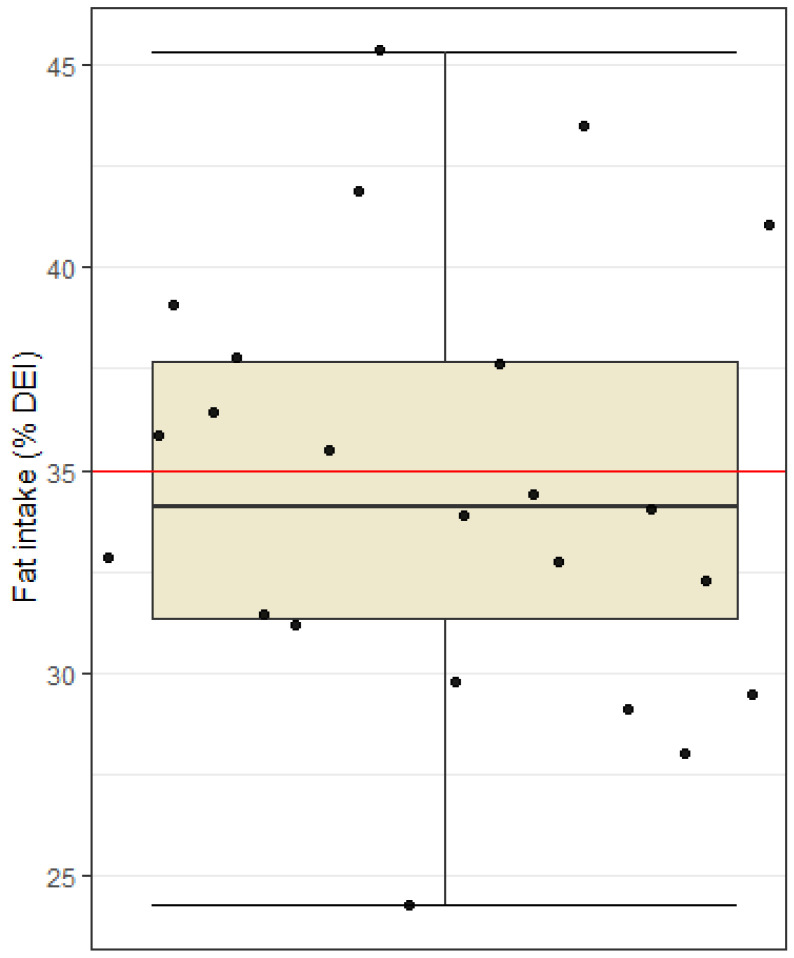
Fat intake among football players. Dots represent individual participants (*n* = 23). Red line indicates upper limit of recommended intake (35% of daily energy intake).

**Figure 5 nutrients-15-00082-f005:**
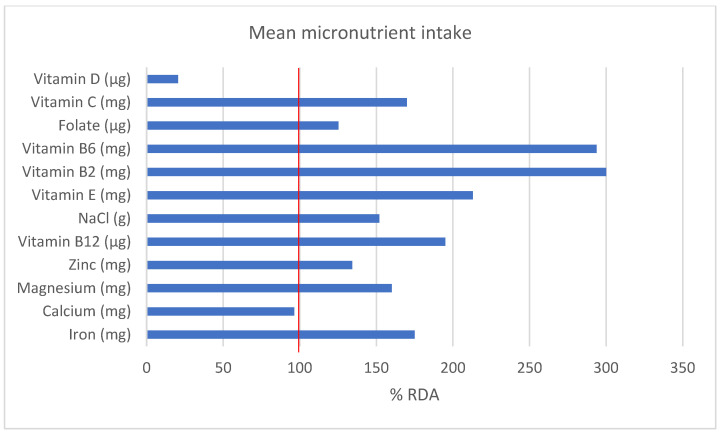
Intake of micronutrients among football players (*n* = 23), presented as a percentage of the recommended daily intake (RDA).

**Figure 6 nutrients-15-00082-f006:**
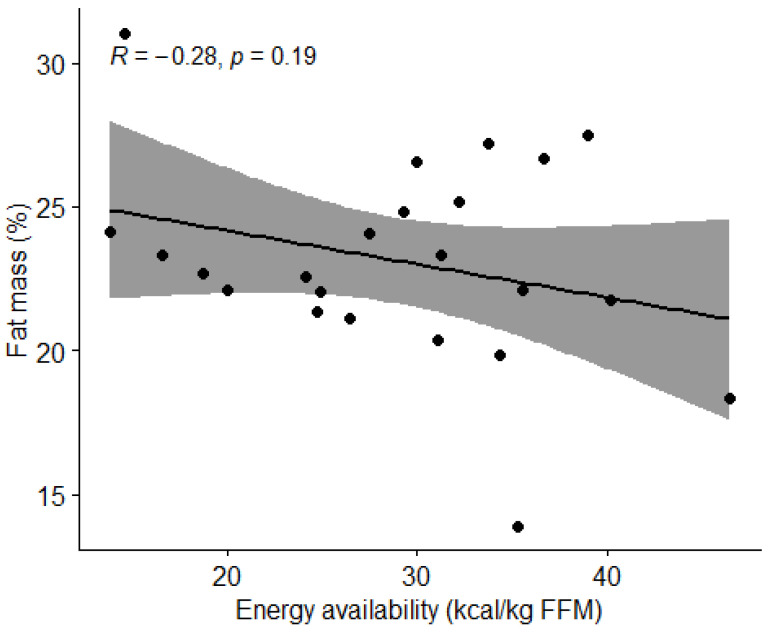
Correlation between energy availability and body fat percentage in football players. Dots represent individual participants (*n* = 23). Grey shaded area indicates 95% confidence interval (CI).

**Figure 7 nutrients-15-00082-f007:**
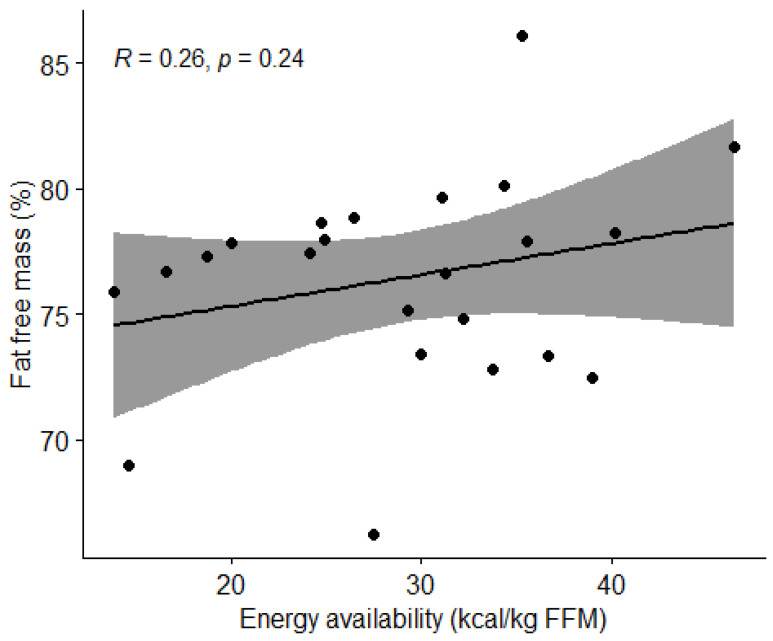
Correlation between energy availability and percentage of lean body mass in football players. Dots represent individual participants (*n =* 23). Grey shaded area indicates 95% CI.

**Figure 8 nutrients-15-00082-f008:**
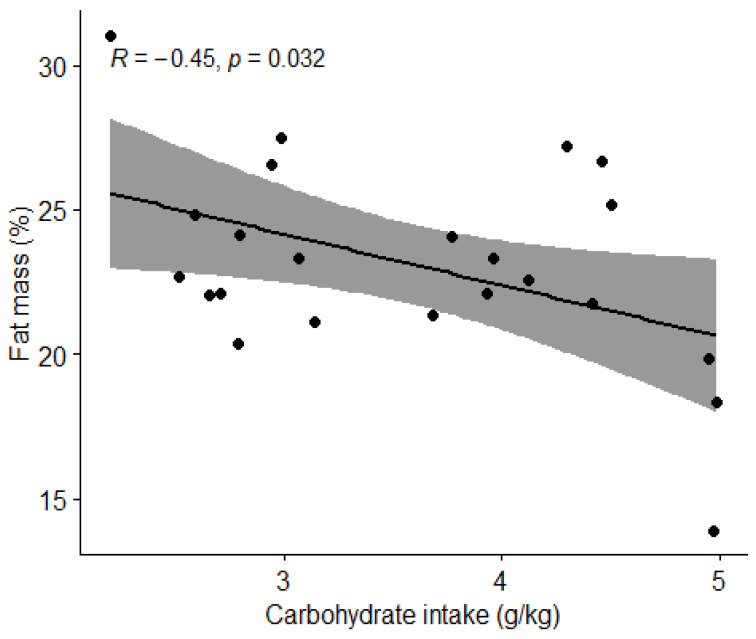
Correlation between carbohydrate intake and body fat percentage in football players. Dots represent individual participants (*n* = 23). Grey shaded area indicates 95% CI.

**Figure 9 nutrients-15-00082-f009:**
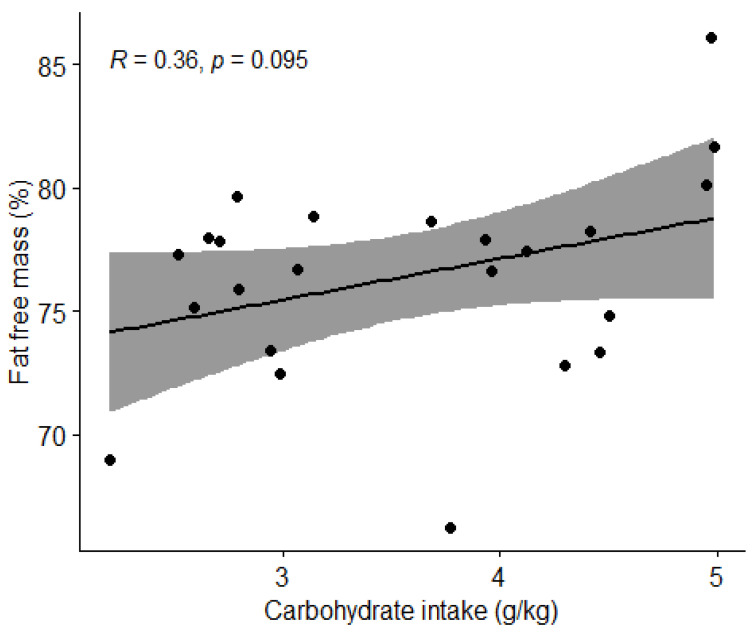
Correlation between carbohydrate intake and percentage of lean body mass in football players. Dots represent individual participants (*n* = 23). Grey shaded area indicates 95% CI.

**Figure 10 nutrients-15-00082-f010:**
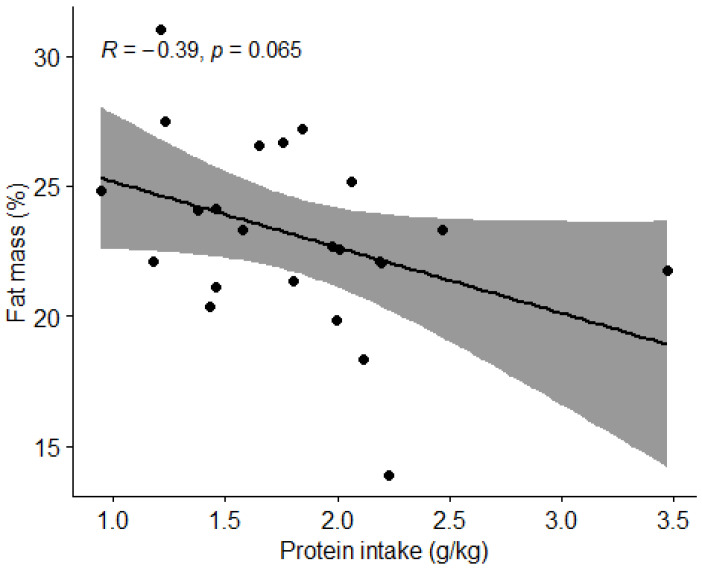
Correlation between protein intake and body fat percentage in football players. Dots represent individual participants (*n* = 23). Grey shaded area indicates 95% CI.

**Figure 11 nutrients-15-00082-f011:**
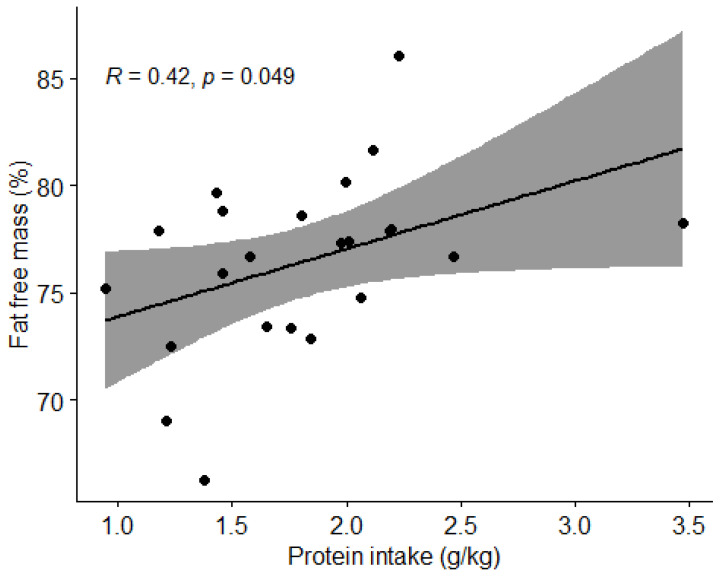
Correlation between protein intake and percentage of lean body mass in football players. Dots represent individual participants (*n* = 23). Grey shaded area indicates 95% CI.

**Figure 12 nutrients-15-00082-f012:**
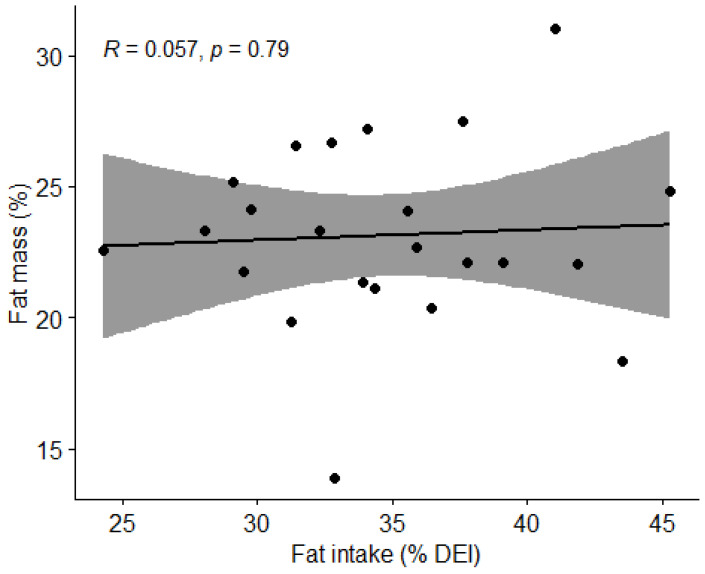
Correlation between fat percentage and body fat percentage in football players. Dots represent individual participants (*n* = 23). Grey shaded area indicates 95% CI.

**Figure 13 nutrients-15-00082-f013:**
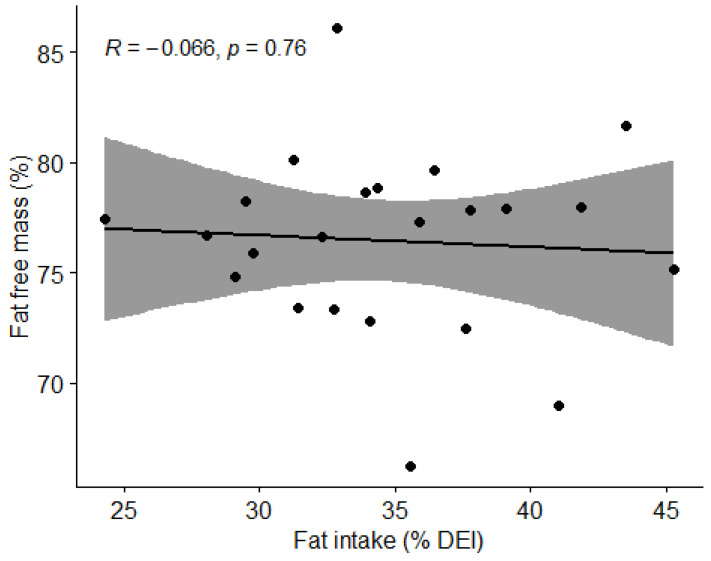
Correlation between percentage of fat and percentage of lean body mass in football players. Dots represent individual participants (*n* = 23). Grey shaded area indicates 95% CI.

**Figure 14 nutrients-15-00082-f014:**
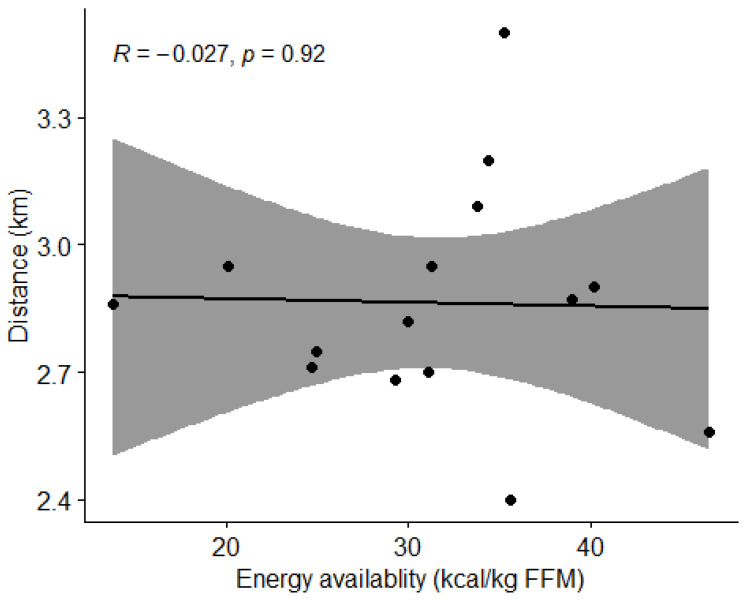
Correlation between energy availability and distance covered by subjects in the Cooper test. Dots represent individual participants (*n* = 15). Grey shaded area indicates 95% CI.

**Figure 15 nutrients-15-00082-f015:**
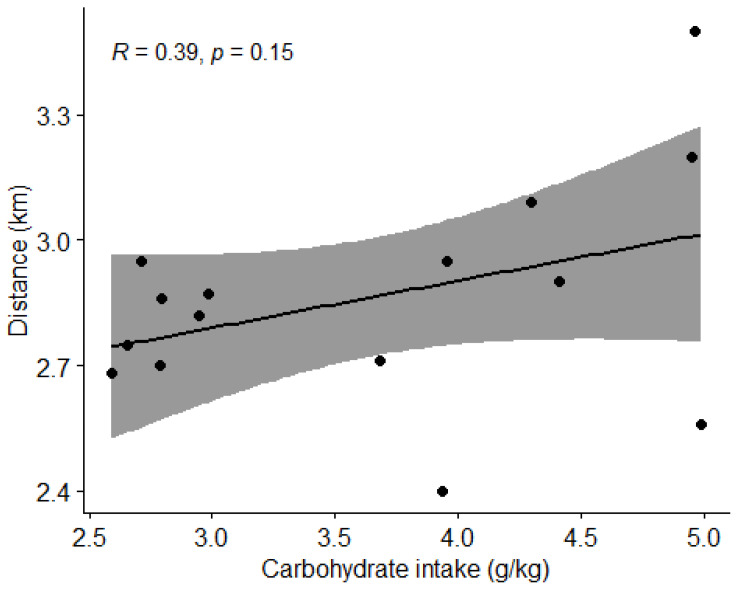
Correlation between carbohydrate intake and distance covered by subjects in the Cooper test. Dots represent individual participants (*n* = 15). Grey shaded area indicates 95% CI.

**Figure 16 nutrients-15-00082-f016:**
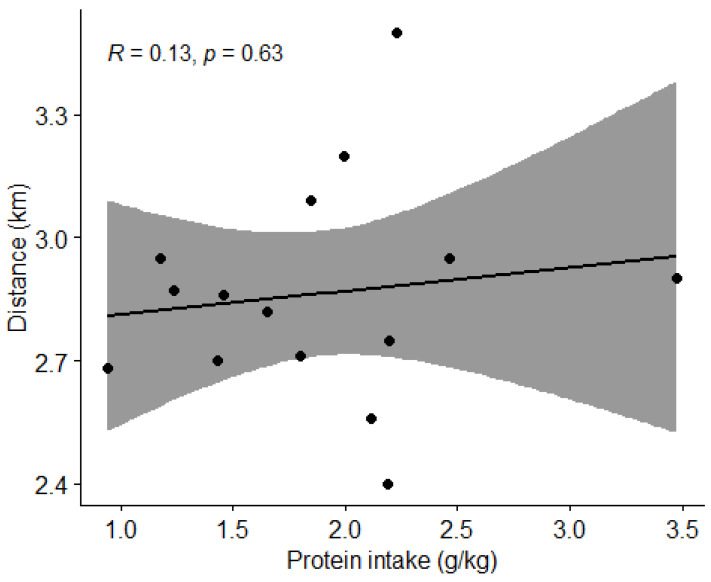
Correlation between protein intake and distance covered by subjects in the Cooper test. Grey shaded area indicates 95% CI. Dots represent individual participants (*n* = 15).

**Figure 17 nutrients-15-00082-f017:**
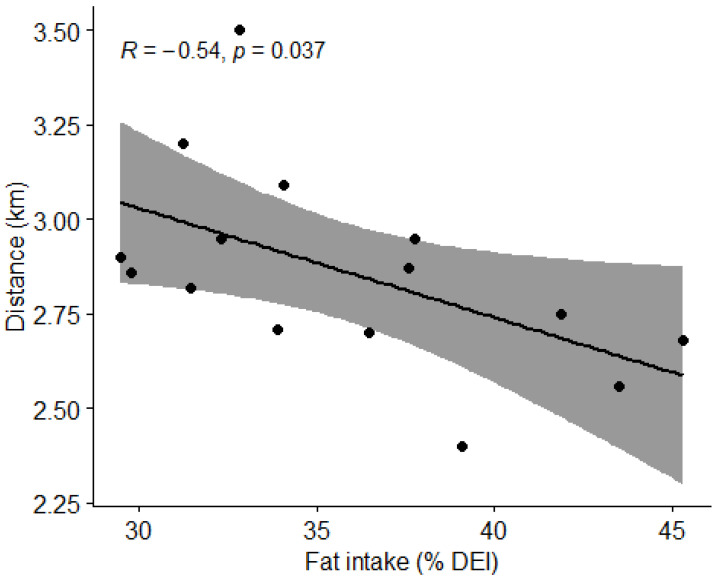
Correlation between the percentage of fat in the diet and distance covered by subjects in the Cooper test. Dots represent individual participants (*n* = 15). Grey shaded area indicates 95% CI.

**Figure 18 nutrients-15-00082-f018:**
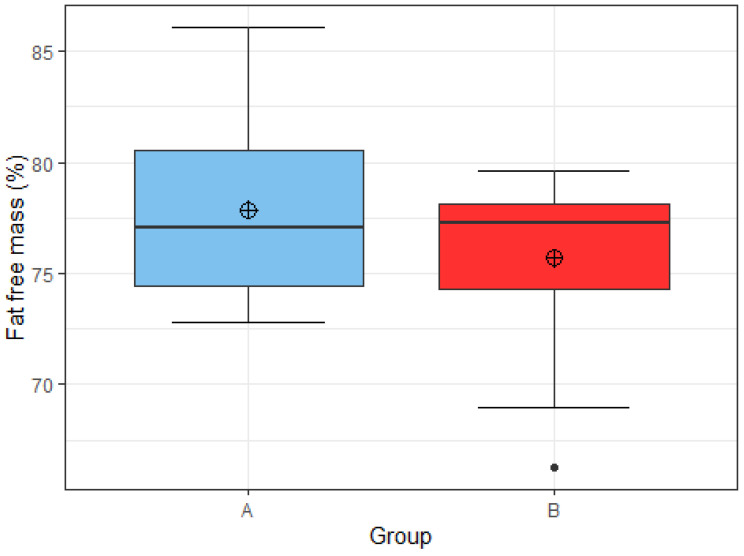
Distribution of percentage of lean body mass by group A (*n* = 8) and group B (*n* = 15). Group A: appropriate intake of protein and carbohydrate. Group B: inappropriate intake of protein or carbohydrate.

**Figure 19 nutrients-15-00082-f019:**
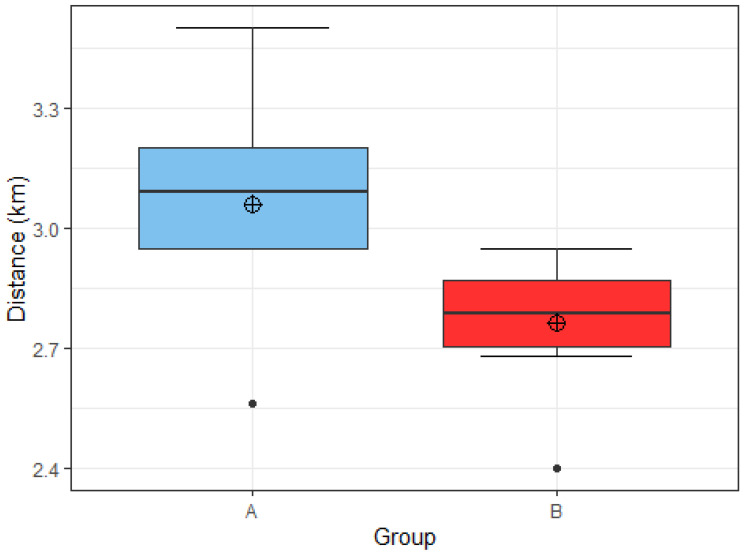
Distribution of distance covered in the Cooper test by group A (*n* = 5) and group B (*n* = 10). Group A: appropriate intake of protein and carbohydrate. Group B: inappropriate intake of protein or carbohydrate.

**Figure 20 nutrients-15-00082-f020:**
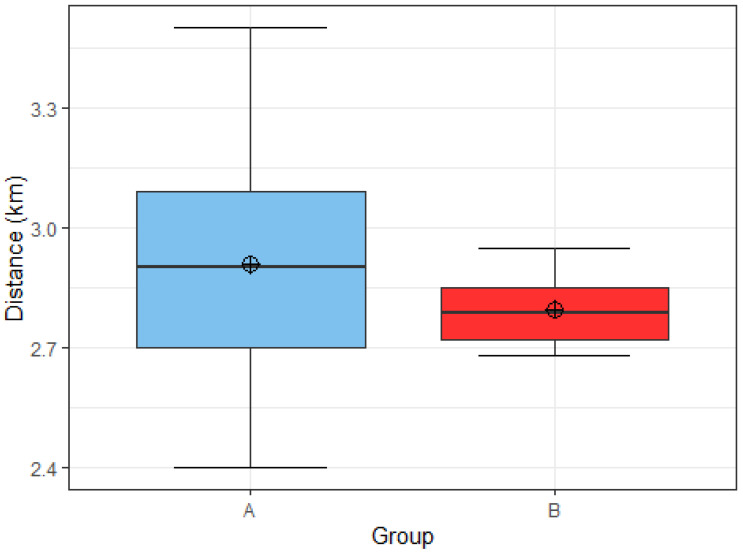
Distribution of distance covered in the Cooper test by group A (*n* = 9) and group B (*n* = 6). Group A: energy availability ≥30 kcal/kg FFM. Group B: energy availability <30 kcal/kg FFM.

**Table 1 nutrients-15-00082-t001:** Participant characteristics.

Variable	Mean ± SD	Range
Age (year)	24 ± 3.4	19–31
Height (cm)	182 ± 6.2	169–192
Body mass (kg)	78 ± 7.4	60.5–88
BMI (kg/m^2^)	23.4 ± 1.2	20.5–25.2
FFM (kg)	59.4 ± 4.8	49.4–66.7
FFM (%)	76.4 ± 4.1	66.3–86.1
FM (kg)	18.2 ± 3.8	8.7–24.8
MET	23.1 ± 3.5	13.9–31.0
EEE (kcal)	12.8 ± 4.5	0.6–15.1

SD—standard deviation; BMI—body mass index; FFM—fat free mass; FM—fat mass; MET—metabolic equivalent of activity; EEE—exercise energy expenditure.

**Table 2 nutrients-15-00082-t002:** Nutritional intake of subjects.

Variable	Mean ± SD	Range
Energy intake (kcal)	2694 ± 494	NA
Energy availability (kcal/kg FFM)	29.0 ± 8.5	≥40
Carbohydrate intake (g/kg BM)	3.6 ± 0.9	4–8
Protein intake (g/kg BM)	1.8 ± 0.5	1.6–2.2
Fat intake (% daily energy intake)	34.7 ± 5.2	<35
Saturated fat intake (% daily energy intake)	10.5 ± 3.8	<10
Fiber intake (g)	27.4 ± 10.2	>30

NA—not applicable; SD—standard deviation; BM—body mass; FFM—fat free mass; BMI—body mass index; MET—metabolic equivalent of activity; EEE—exercise energy expenditure.

**Table 3 nutrients-15-00082-t003:** Micronutrient intake of subjects.

Variable	Mean ± SD	RDA [7,21]
Iron (mg)	17.5 ± 11.8	10
Calcium (mg)	964.2 ± 298.8	1000 oz. 1500
Magnesium (mg)	560.5 ± 175.4	350–400
Zinc (mg)	18.8 ± 7.5	11–16
Vitamin C (mg)	186.9 ± 216.8	110
Vitamin B_2_ (mg)	3.9 ± 6.0	1.3
Vitamin B_6_ (mg)	4.7 ± 4.4	1.6
Folate (µg)	375.6 ± 227.9	300
Vitamin B_12_ (µg)	7.8 ± 3.6	4
Vitamin E (mg)	27.7 ± 36.3	13
Vitamin D (µg)	4.1 ± 3.1	20
NaCl (g)	7.6 ± 2.4	<5

SD—standard deviation; RDA—recommended daily intake; NaCl—sodium chloride.

## Data Availability

Not applicable.

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
