# Peer review of "Dietary Intake, Body Composition and Performance of Professional Football Athletes in Slovenia"

_nutrients, 2022, doi:10.3390/nu15010082_

Round 1

Reviewer 1 Report

Dear Authors,

The manuscript presents a theme of significant scientific relevance. However, I have some comments.

 1. In line 11, "The aim of this study was." I suggest "This study aimed."

2. In the abstract, the sentence showed fragmented (lines 22 to 25).

3. What is the meaning of FFM? (line 74)

4. About the Medical Ethics Commission of the Republic of Slovenia. What is the registration number referring to experiments with humans? (line 89).

5. How do the authors explain many out layers in Figures 1, 3, and 4?

Author Response

Dear reviewer,

Thank you for your time and comments. They are greatly appreciated!

Comments 1, 2, 3, and 4 have been properly addressed and corrected in the text.

As for comment number 5 regarding outliers in Figures 1, 3, and 4. These findings might be explained by the fact that subjects included in the study come from very different social and financial backgrounds, upbringings, and different parts of the country with varying levels of baseline nutritional knowledge and nutritional awareness.

For instance, certain subjects showed little to no interest in nutrition despite the fact they are professional football players and football was their main and/or only source of income. On the other hand, other players were aware of the impact that nutrition plays on their performance and health. These factors might explain to some degree the fact that certain individuals were mindful to eat enough energy and the right amount of macronutrients according to their knowledge.

Another thing to take into consideration is that some of these players were getting their nutritional information from untrustworthy sources, i. e. social media. From talking to these individuals, it was apparent their nutritional knowledge was skewed. For example, two individuals eat approximately 3.5 g of protein/kg BW because they were under the impression that protein is extremely important for their performance, which it is to a certain extent, however, put too much emphasis on this macronutrient whilst eating too few calories from carbohydrates and energy overall. There was also a prevailing thought that carbohydrates lead to fat gain as discussed in the text.

Finally, participants eat quite differently despite having two meals provided for them at the football club. At breakfast, they were given a wide variety of different foods to choose from. The foods provided were the following: eggs, cereal, oats, a wide variety of fruits, regular and Greek yogurt, milk, butter, marmalade, bread, coffee, fruit juice, and different types of salami and cheese. Subjects could choose from any of these foods and quantity of these foods of their own volition. Consequently, subjects eat quite differently at this meal. The other meal provided was lunch, the most uniform meal across all subjects, differing slightly only in quantities of carbohydrates eaten in the form of pasta, rice, potatoes, etc. However, the most considerable discrepancies seen in nutritional intakes were seen after lunch when participants had to provide their own meals. Afternoon snacks and dinners varied greatly between individuals, and this could explain why energy and macronutrient intakes (figures 1 to 4) were recorded very differently. This part has been added to the discussion section to further elucidate the findings (lines 571 to 583).

We hope we have sufficiently addressed the comments provided.

Kind regards,

Matjaž Macuh

Reviewer 2 Report

This paper describes the relationship between food intake, body composition, and the exercise capacity of football players from a pro football club in Slovenia. However, as indicated in the discussion, the number of subjects is very small, and the food intake expected to eat similar meals belonging to the same club is recorded very differently, making it difficult to accept the measured results as a sample. Therefore, the statements in the conclusions should be revised to reflect the assumptions based on the results.

Author Response

Dear reviewer,

Thank you for your time and comments. They are greatly appreciated!

Regarding the differences in food records, these findings could be explained by many factors. Firstly, subjects included in the study come from very different social and financial backgrounds, upbringings, and different parts of the country with varying levels of baseline nutritional knowledge and nutritional awareness.

For instance, certain subjects showed little to no interest in nutrition despite the fact they are professional football players and football was their main and/or only source of income. On the other hand, other players were aware of the impact that nutrition plays on their performance and health. These factors might explain to some degree the fact that certain individuals were mindful to eat enough energy and the right amount of macronutrients according to their knowledge.

Another thing to take into consideration is that some of these players were getting their nutritional information from untrustworthy sources, i. e. social media. From talking to these individuals, it was apparent their nutritional knowledge was skewed. For example, two individuals eat approximately 3.5 g of protein/kg BW because they were under the impression that protein is extremely important for their performance, which it is to a certain extent, however, put too much emphasis on this macronutrient whilst eating too few calories from carbohydrates and energy overall. There was also a prevailing thought that carbohydrates lead to fat gain as discussed in the text.

Finally, participants eat quite differently despite having two meals provided for them at the football club. At breakfast, they were given a wide variety of different foods to choose from. The foods provided were the following: eggs, cereal, oats, a wide variety of fruits, regular and Greek yogurt, milk, butter, marmalade, bread, coffee, fruit juice, and different types of salami and cheese. Subjects could choose from any of these foods and quantity of these foods of their own volition. Consequently, subjects eat quite differently at this meal. The other meal provided was lunch, the most uniform meal across all subjects, differing slightly only in quantities of carbohydrates eaten in the form of pasta, rice, potatoes, etc. However, the most considerable discrepancies seen in nutritional intakes were seen after lunch when participants had to provide their own meals. Afternoon snacks and dinners varied greatly between individuals, and this could explain why energy and macronutrient intakes (figures 1 to 4) were recorded very differently. This part has been added to the discussion section to further elucidate the findings (lines 571 to 583).

Unfortunately, all the highlighted problems are very common when working with individuals in the field, especially in Slovenia where nutritional awareness and knowledge of athletes in general (not just football players) could be greatly improved.

Regarding the sample size included a greater sample size would be ideal, however, as researchers we know this is not always possible. We tried to include every single player from the football club, and most of the players were willing to cooperate. We must respect the fact that the football club included was even willing to cooperate to the extent they were and let us (the researchers) interfere, even if ever so slightly, into their regular routine whilst having to prepare for matches. That was greatly appreciated. Lastly, we believe that for sports nutrition research the number of subjects included in this study is not uncommon and sufficient to draw the conclusions presented whilst taking into account the limitations of the study discussed in greater detail in the main text.

We hope we have sufficiently addressed the comments provided.

Kind regards,

Matjaž Macuh

Round 2

Reviewer 2 Report

Conclusions should be free from theorizing and explained in terms of results. Therefore, the discussion on sports nutrition described in the conclusions lacks grounds to clarify the results of this study.

Author Response

Dear reviewer,

Thank you for your time and comments!

We have changed the conclusion section so that it is only explained in terms of results.

We hope we have sufficiently addressed the comments provided.

Kind regards,

Matjaž Macuh